# Natural Killer Cells and Cytotoxic T Cells: Complementary Partners against Microorganisms and Cancer

**DOI:** 10.3390/microorganisms12010230

**Published:** 2024-01-22

**Authors:** Aristo Vojdani, Sadi Koksoy, Elroy Vojdani, Mark Engelman, Carina Benzvi, Aaron Lerner

**Affiliations:** 1Immunosciences Laboratory, Inc., Los Angeles, CA 90035, USA; 2Cyrex Laboratories, LLC, Phoenix, AZ 85034, USA; s.koksoy@cyrexlabs.com (S.K.); engelman.m-ne@cyrexlabs.com (M.E.); 3Regenera Medical, Los Angeles, CA 90025, USA; drv@regeneramedical.com; 4Chaim Sheba Medical Center, The Zabludowicz Research Center for Autoimmune Diseases, Ramat Gan 52621, Israel; carina.ben.zvi@gmail.com (C.B.); aaronlerner1948@gmail.com (A.L.)

**Keywords:** NK subpopulation, natural killer cell, cytotoxic T cell, microorganism, granules, innate immunity, adaptive immunity, immunosenescence

## Abstract

Natural killer (NK) cells and cytotoxic T (CD8^+^) cells are two of the most important types of immune cells in our body, protecting it from deadly invaders. While the NK cell is part of the innate immune system, the CD8^+^ cell is one of the major components of adaptive immunity. Still, these two very different types of cells share the most important function of destroying pathogen-infected and tumorous cells by releasing cytotoxic granules that promote proteolytic cleavage of harmful cells, leading to apoptosis. In this review, we look not only at NK and CD8^+^ T cells but also pay particular attention to their different subpopulations, the immune defenders that include the CD56^+^CD16^dim^, CD56^dim^CD16^+^, CD57^+^, and CD57^+^CD16^+^ NK cells, the NKT, CD57^+^CD8^+^, and KIR^+^CD8^+^ T cells, and ILCs. We examine all these cells in relation to their role in the protection of the body against different microorganisms and cancer, with an emphasis on their mechanisms and their clinical importance. Overall, close collaboration between NK cells and CD8^+^ T cells may play an important role in immune function and disease pathogenesis. The knowledge of how these immune cells interact in defending the body against pathogens and cancers may help us find ways to optimize their defensive and healing capabilities with methods that can be clinically applied.

## 1. Introduction

More than 50 years ago, lymphocytes were classified as either T cells (T for thymus) or B cells (B for bone), but in 1973, Greenberg et al. first described a “null” killer cell that was neither a T nor B cell, which could efficiently kill antibody-coated cells [1,2,3]. A year later, Greenberg and Playfair reported on an unusual type of “spontaneous” cytotoxicity in which these “null” lymphocytes could kill target cells without prior immunization and without an antibody [4]. Another year later, Keissling et al. determined that the killer cell was genetically regulated and killed many types of tumors. They named this mysterious and exciting new effector cell the “natural” killer (NK) cell for its naturally spontaneous cytotoxicity [5].

In the same year, 1975, the phenotypic and functional separation of CD8 and CD4 T cells was established [6,7,8]. However, it would still be many years before it would really be understood how important CD8^+^ T cells were to immunity and how they worked. Almost 15 years after the discovery that CD8 was associated with the lytic fraction of Thy-1-bearing lymphocytes [8], it came to be regarded as no longer just a marker, but instead, closely tied to target recognition [9]. Fast forward almost two decades, and we now know the CD8^+^ T cell to be the natural killer cell’s important partner in immune defense. And we cannot discuss NK cells without mentioning a third group of immune cells. In 2008, more than 130 years after the identification of lymphoid and myeloid cells, and more than 40 years after the identification of T and B lymphocytes, a series of studies from 12 independent laboratories around the world unveiled a new, unexpected group of players within the immune system, specifically, the innate lymphoid cells (ILCs), an important family of developmentally related cells that are involved in immunity and in tissue development and remodeling [10]. ILCs were shown to have subsets remarkably similar to Th cell subsets, and in 2013, Spits et al. [11,12] proposed a uniform nomenclature that divides ILCs into three groups. Group 1 ILCs are defined by the production of the signature cytokine IFN-γ and are typified by their most famous member, the NK cell; yes, the NK cell is actually an ILC1. Group 2 ILCs require IL-7 for their development and produce Th2-cell-associated cytokines. And Group 3 ILCs are defined by their capacity to produce the cytokines IL-17A and/or IL-22, depending on the transcription factor RORγt for their development and function, much like Th17 cells.

## 2. Natural Killer (NK) Cells and Their Subpopulations

Natural killer (NK) cells are a vital part of our immune system’s defense against pathogens and cancer. They originate from bone marrow, lymph nodes, and liver in the form of NK cell precursors, which differentiate into NK cells and no other lineages [13]. The cytokine IL-15 is essential for NK cell development and homeostasis. While NK cells are considered major components of the immune system in protecting the body against pathogens and cancer, like macrophages, neutrophils, and the like, they are neither considered B nor T cells and are best described as innate cytotoxic lymphocytes with adaptive immune features, so that they are now considered a bridge between innate and adaptive immunity [14,15,16,17,18,19]. They constitute about 5–15% of all lymphocytes circulating in the blood, which makes them the third largest lymphocyte population, after T and B cells. Humans with deficient or dysfunctional NK cells are susceptible to infections that can potentially be life-threatening [20]. NK cells contribute to host defense in several ways, both directly and indirectly [21,22].

What distinguishes NK cells from other immune cells is their inherent ability to respond rapidly to pathogens without prior activation. Unlike many immune cells that require a specific stimulus or activation signal to initiate a response, NK cells are primed to detect and attack invading pathogens immediately upon their entry into the body, without any priming or prior activation by other cells. Instead, NK cells basically have a form of friend-or-foe identification system. For the NK cell, the friend identifier is the major histocompatibility complex I (MHC-I) molecule on the suspect cell’s surface [23,24]. If the NK cell finds an MHC-I molecule on the surface of a suspect cell, it will not destroy that cell. If an NK cell does not find an intact MHC-I on a target cell, it will go into kill mode and destroy that cell. However, there is a third scenario wherein a target cell still has an MHC-I molecule but also expresses a ligand, such as a viral antigen, that identifies it as a harmful cell. In this case, the ligand overrides the inhibitory “friend” signal from the MHC-I, and the NK cell goes into kill mode on the target cell [23,24].

This ability to tell friend from foe is part of what makes NK cells extremely efficient at finding and killing viruses, cancers, and the cells infected by them. This is also what makes NK cells so good at their job as part of the body’s first line of defense against pathogens. NK cells also kill bacteria, fungi, and parasites that invade the human body, either killing these microbes directly or the cells that they have infected [25,26,27,28,29]. Low numbers of NK cells are associated with increased prevalence of fungal infection, and there is a correlation between reduced NK cell cytotoxic activity and fungal infections [30]. NK cells have been shown to be active and cytotoxic against bacteria such as *Salmonella typhimurium* [31], *Shigella flexneri* [32], *Mycobacterium tuberculosis* [33], *Staphylococcus aureus* [34], *Listeria monocytogenes* [35], and *Pseudomonas aeruginosa* [36], and against mold and fungi such as *Candida albicans* [37,38], *Cryptococcus neoformans* [38,39], *Aspergillus fumigatus* [40,41], *Paracoccidioides braziliensis* [42], *Coccidioides immitis* [43], *Bacillus anthracis* [44], and *Pneumocystis murina* [45].

Studying the various ways in which NK cells make use of the many different components of cellular toxicity may provide valuable information about defending against harmful microorganisms [46].

The immunophenotyping of the different NK and cytotoxic T-cell subpopulations is made possible because the surfaces of these cells express special molecules called cellular differentiation or cluster differentiation (CD) markers. Included in these markers are the receptors and ligands that enable cytotoxic protective immune cells to tell normal, harmless, or host cells from diseased, harmful, or pathogenic organisms [47]. These CD markers have been identified and numbered and are used to distinguish the different NK-cell and T-cell subpopulations from each other.

Among the different receptors found on their surfaces, NK cells carry three major CD markers: CD16, CD56, and CD57. Based on the amount, strength, and concentration of the markers found on their cell membrane, NK cells can be classified into three basic subsets [26,48].

### 2.1. What Are CD56^+^CD16^dim^ NK Cells?

CD56^bright^CD16^neg/dim^ or CD56^+^CD16^−^ NK cells make up about 10% of NK cells and are considered immature. These NK cells are found mainly in the lymphoid tissues. Despite the fact that they are only weakly cytotoxic, they are called regulatory NK cells because they are abundant producers of cytokines and chemokines such as interferon-gamma (IFN-γ), tumor necrosis factor beta (TNF-β), interleukin-10 (IL-10), and IL-13, giving them immuno-regulatory properties and a role in shaping the adaptive immune response. This cytokine-producing ability has been found to be negatively affected in people with myeloma. Phenotypic alterations of the CD56^+^ immune cell fraction have been reported in patients with various infectious autoimmune or malignant diseases [26,27,28].

### 2.2. What Are CD56^dim^CD16^+^ NK Cells?

CD56^dim^CD16^+^ or CD56^−^CD16^+^ NK cells constitute about 90% of circulating NK cells and are considered mature. These are cytotoxic NK cells that are involved in the cytotoxicity of viral-infected cells and tumor cells. They are also involved in cytokine production. Most NK cells express the activating CD16 receptor. They also express Fc-gamma receptors that engage the Fc region of IgG [49]. The CD16 receptor lacks the ability to signal on its own; therefore, it requires an adaptor molecule that contains an immunoreceptor tyrosine-based activating motif (ITAM) to mediate signaling cascades. In NK cells, cross-linking of the CD16 receptor results in phosphorylation of the ITAM(s) that initiate signaling cascades, which ultimately lead to the release of cytotoxic granules containing granzymes and perforin, a process known as antibody-dependent cellular cytotoxicity (ADCC), leading to killing of target cells. When an antibody binds to a receptor on a tumor cell, the CD56^−^CD16^+^ NK cell binds to the antibody, allowing the NK cell to release its cytotoxic granules to kill the cell. Likewise, if an anti-EBV antibody binds to a receptor on an EBV-infected cell, the CD56^−^CD16^+^ NK cell can bind to that antibody, induce the release of granules, and kill the virus-infected cell [49].

### 2.3. What Are CD57^+^ Cells?

CD57 is a marker protein found on the surface of a subset of NK cells. They are a highly mature or developed, discrete, functionally distinct subpopulation of CD16^+^ NK cells that are involved in immunoprotecting against pathogens and other environmental factors. Also called memory NK cells, they display high cytotoxic potential and are highly differentiated long-lived cells that are thought to be more efficient at killing their targets compared to other NK-cell subpopulations. CD57^+^ NK cells have become a matter of recent interest because they are now believed to play a unique role in immune function and have been implicated in a number of health conditions [48,49,50,51,52,53,54,55,56]. The CD57 marker is also detected on neural cells, in which it acts as an adhesion molecule, and thus may be involved in the communications between the nervous system and the immune system [57]. CD57^+^ NK cells are consistently associated with better outcomes in cancer and autoimmune disease. Low levels of CD57^+^ are associated with lower overall survival in cancer [58,59,60,61,62,63,64,65,66,67,68,69,70,71].

CD57 is a very useful marker of NK-cell maturation. Studies show that progression from CD56^bright^ to CD56^dim^CD57^−^ to CD56^dim^CD57^+^ reflects a maturation pathway for NK cells [16]. Acquisition of CD57 means a higher cytotoxic capacity and greater responsiveness to signaling via CD16 and natural cytotoxic receptors. CD57^+^ cells identify the final stage of NK cell maturation, and their numbers increase with aging [48,51,72,73]. The numbers of CD57^+^ NK cells also greatly increase in response to chronic exposure to antigens originating from tumor cells, bacteria, and viruses, particularly with EBV, CMV, HIV, hepatitis C, and more. Increases in the detectable numbers of CD57^+^ NK cells in the blood are also associated with autoimmune disease [48,60,61].

In different autoimmune diseases, CD57^+^ NK cells fulfill an immunoregulatory role with their ability to delete autoreactive T cells that are chronically activated by viral antigens. Generally, increases in the population of autoreactive CD57^+^ cells are associated with more severe autoimmune diseases such as Wegener’s granulomatosis [74], multiple sclerosis (MS) [75], type 1 diabetes [76], Graves’ disease [77], and rheumatoid arthritis [78]. However, there are some instances where certain autoimmune diseases are consistently associated with reduced frequencies or absolute numbers of circulating CD57^+^ NK cells and/or impaired cytotoxicity [79,80,81,82,83,84,85,86,87]. In several autoimmune diseases such as atopic dermatitis [88,89], Sjögren’s syndrome [90], IgA nephropathy [91], psoriasis [92], and alopecia areata [93], a reduction in CD57^+^ NK cells in peripheral blood was reported in comparison to controls.

CD57^+^ NK cells are very important in early pregnancy failure. In fact, NK cell counts were shown to be greater in a subgroup of patients who suffered recurrent pregnancy failure (3.42% ± 2.15) compared to controls (2.14% ± 1.42) [51].

In relation to many pathogens that induce or cause an increase in CD57^+^ cells, in 2001, it was reported that there was a reduction in the blood levels of CD57^+^ NK cells in patients with chronic Lyme disease in comparison to those with acute Lyme disease and uninfected individuals [94,95]. However, a 2009 study found no differences between the NK cell counts of patients with chronic Lyme disease and those of controls [96]. In agreement with this article, many authorities such as the CDC [97], North American medical experts [98], European science organizations [99], and the Royal College of Pathologists of Australasia [100] have explicitly warned against the use of CD57 and other unvalidated tests in the diagnosis of Lyme disease. Thus, CD57^+^ cells should be measured not for the diagnosis of Lyme disease but for the indication of chronic exposure to antigens that originate from bacteria, viruses, tumor cells, and neoantigens formed in the body. The counting of CD57^+^ cells is also recommended for patients with various autoimmune diseases, as the majority of them are induced by environmental factors [48].

### 2.4. What Are CD57^+^CD16^+^ NK Cells?

CD57^+^CD16^+^ cells are a subpopulation of CD56^dim^CD16^+^ NK cells that carry CD57 markers of their surfaces. Cells that carry both CD57 and CD16 markers are highly cytotoxic.

CD57^+^CD16^+^ cells contain the cytolytic enzymes granzyme A, granzyme B, and perforin in their granules, enabling them to kill cancer cells and virus-infected cells with great efficiency. These cells are considered the most potent cells for combating acute and chronic viral infection [48,50].

Elevation in CD57^+^CD16^+^ cells is an indication of persistent antigenic stimulation, especially from viruses. Human CMV is one of the clearest examples of infection driving NK cell differentiation, particularly driving the expansion of NKG2C^+^ NK cells, which preferentially acquire CD57. Mature CD57^+^ NK cells expand as a consequence of lifetime exposure to infections, including HBV, HCV, EBV, CMV, hantavirus, and HIV. Their increase is an indication of ongoing or previous viral infection. Elevation in the levels of CD57^+^ NK cells observed in patients with chronic viral infection correlates with slower disease progression [48,50].

Compared to healthy controls, in patients with severe COVID-19, high frequencies of CD57^+^CD16^+^ were detected despite the absolute number of these cells being decreased due to low WBC and low lymphocyte counts. Individuals with long COVID showed significantly increased levels of functional memory cells with high antiviral cytotoxic activity, including NK cells with the CD56^+^CD57^+^NKG2C^+^ phenotype. The hypothesis that a persistent memory cytotoxic response against SARS-CoV-2 could be the cause of long COVID is supported by NK cells expressing both memory (CD57) and activation (NKG2C) markers that did not go away even after the initial SARS-CoV-2 infection was cleared [101,102,103,104,105].

In autoimmune diseases, reduced frequencies or absolute numbers of circulating CD57^+^ NK cells and/or impaired NK-cell cytotoxicity have consistently been observed. This supports the regulatory role of cytotoxic CD57^+^ NK cells in preventing or suppressing autoimmune disease. For example, the blood level of CD57^+^CD16^+^ cells is depleted in inflammatory and some autoimmune disorders [75,76,77,78,79,80,81,82,83,84,85,86,87,88,89,90,91,92,93,94,95], such as psoriasis [85], atopic dermatitis [88,89], rheumatoid arthritis [81], and multiple sclerosis [79,80]. However, an increase in the number of CD57^+^ CD16^+^ cells has been shown in tissues, such as the skin, joints, and pancreas, of affected individuals. Increased oxidative stress in patients with SLE may have caused preferential apoptosis of mature CD56^dim^CD57^+^ NK cells, perhaps contributing to the impairment of the ability to eliminate pathogenic CD4^+^ T cells seen in SLE [53,83,90,106]. This impairment in the number and cytotoxic capacity of CD57^+^CD16^+^ NK cells may be due to various mechanisms that different pathogens employ in the induction of some autoimmune diseases.

## 3. Cytotoxic CD8^+^ T Cells or Cytotoxic T Lymphocytes (CD8^+^, CTL, TC)

CD8^+^ T cells, like NK cells, originate in the bone marrow. However, what makes these cells different is that they migrate to the thymus and develop into maturity there, which is where the “T” in their name comes from [9,107]. They include cytotoxic T cells, which are the main cellular warriors of the adaptive immune system, specializing in killing virally infected or malignant cells. They are the adaptive counterparts of the innate NK cells [108]. They are called cytotoxic lymphocytes because they express cytotoxic molecules such as granzyme A, granzyme B, and perforin. They also produce IFN-γ and TNF-α. These functions and properties together enable them to kill viral-infected cells and cancer cells with a high degree of efficiency. They are considered the most potent cells for stopping the spread of pathogens and tumor cells [109].

In order for cytotoxic CD8^+^ T cells to kill their pathogenic target cells, they first have to be activated or primed by antigen-presenting cells, which give them little pieces of the pathogen’s material so that the CD8^+^ T cells can identify their targets. Cytotoxic CD8^+^ T cells are very efficient in the killing pathogens, particularly viruses that live in the cell. Cells harboring such viruses express on their surface viral peptides bound to major histocompatibility complex I (MHC-I) molecules that are detectable by the cytotoxic CD8^+^ T cell [59,108,109,110].

Thus, NK cells and CD8^+^ T cells are a complementary pairing, cytotoxic partners in defeating pathogenic cells. Since NK cells are from the innate immune system and CD8^+^ T cells are from the adaptive immune system, they are, as Rosenberg and Huang call them, parallel and complementary soldiers of immunotherapy [111].

### 3.1. Important CD8^+^ T-Cell Subsets

CD57^+^ CD8^+^ T cells are a critical subpopulation of CD8^+^ T cells that are mediators of adaptive immunity. CD8^+^ is formidable enough alone, but putting CD8^+^ together with CD57^+^ to obtain CD57^+^CD8^+^ is like combining two highly effective tools into one master tool. There is growing evidence that the CD57^+^CD8^+^ T-cell population plays a significant role in various diseases or conditions associated with chronic immune activation [59]. As a result of lifetime exposure to common antigens such as pathogens, neoantigens, and autoantigens, the numbers of CD57^+^CD8^+^ cells increase with age. This is because persistent exposure to these antigens induces the expansion of CD57^+^CD8^+^ cells [112,113,114].

An increase in the CD57^+^CD8^+^ T-cell population is observed in individuals with chronic infections such as EBV, CMV, measles, hepatitis C, parvovirus, HIV, SARS-CoV-2, toxoplasma, and more.

An increase in the numbers of CD57^+^CD8^+^ T cells has been observed in the blood of patients with different malignancies [52,53,54,55,56,57,58,59,60,61,62,63], including melanomas, head and neck cancer, and hemato-oncological diseases.

Quantitative changes in the CD57^+^CD8^+^ T-cell population are observed in different autoimmune diseases, such as MS [79,80], type 1 diabetes [76], Graves’ disease [77], ankylosing spondylitis [59], and rheumatoid arthritis [78,81]. In Graves’ disease [77], ankylosing spondylitis [59], polymyositis [59], dermatomyositis [59,86], and rheumatoid arthritis [78], increased CD57^+^CD8^+^ T cells are also associated with the severity of the disease. In rheumatoid arthritis, after treatment with abatacept, a decrease in CD57^+^CD8^+^ T cells correlated with clinical response [115]. In other autoimmune diseases, including lupus, type 1 diabetes [75], and MS [79,80], decreases in the number of CD57^+^CD8^+^ T cells were observed. It has also been observed that CD8^+^CD57^+^ T cells in tumors lack cytotoxic activity, although the boosting of IL-15 was able to restore the impaired proliferative activity of CD8^+^CD57^+^ T cells in tumors and peripheral blood [116].

And finally, increases in the numbers of CD57^+^CD8^+^ T cells have been observed in chronic pulmonary disease [117], chronic alcoholism [59], organ and bone marrow transplantation [118], and acute physical stress [59].

Another subset of CD8^+^ T cells is composed of KIR^+^CD8^+^ cells, which are present in lymph nodes, spleen, and peripheral blood and make up to 4.5% of total T cells in healthy adults [119]. These cells exert immunosuppressive and immunoregulatory capabilities. KIR^+^CD8^+^ cells exhibit high levels of cytotoxic molecules such as granzyme B and perforin [120]. The population of these cells expands during viral infection, cancer, aging, and many autoimmune disorders [120,121,122,123]. In relation to autoimmune diseases and infections, Li et al. [123] not only found increased frequencies of KIR^+^CD8^+^ in celiac disease and in patients with coronavirus infection but also demonstrated immunosuppression and immunoregulatory function against pathogenic CD4^+^ T cells. They concluded that development of autoimmunity caused by viral infection could be prevented by KIR^+^CD8^+^ regulatory T cells. Furthermore, in a very recent article, Paris-Muñoz et al. discussed the importance of the Helios *IKZF2* gene transcription factor and its influence on KIR^+^CD8^+^ Tregs and other immunosuppressive cells in transfer therapies as a potential treatment for systemic lupus erythematosus (SLE) and other autoimmune disorders [124]. These immunosuppressive and immunoregulatory capabilities of KIR^+^CD8^+^ Tregs and their importance in preventing the viral initiation of autoimmunity through the induction of programmed cell death in autoreactive Th1 and Th17 pathogenic T cells are shown in Figure 1.

### 3.2. What Are CD3^+^CD16^+^CD56^+^ NKT Cells?

CD3^+^CD16^+^CD56^+^ cytotoxic natural killer T cells or NKT cells constitute a unique and very rare subset of CD1d-restricted T cells that serve as a bridge between innate and adaptive immunity because they carry receptors characteristic of both T cells and NK cells. They have major modulating effects on immune responses via the secretion of cytokines. NKT cells are considered important players in tumor immunosurveillance, and the have been shown to play a role in immune protection against microbial pathogens, in different types of cancers, and in the control of autoimmune diseases [125,126].

Th1-like NKT cells can induce an antitumor response, while Th2- and Treg-like NKT-cell subsets facilitate immune escape and tumor progression. An important subset of NKT cells comprises the invariant NKT or iNKT cells. These CD1d-dependent NKT cells express an invariant T-cell receptor alpha chain. While NKT cells generally come to be exhausted in advanced cancer, iNKT cells actually increase in activation and effector function within the breast tumor microenvironment [127,128]. The number of circulatory NKT cells is significantly decreased in patients with different cancers compared to healthy controls. Low NKT-cell numbers correlated with poor clinical outcomes in patients with some kinds of cancer [129]. Metelitsa et al. analyzed 98 untreated primary neuroblastomas from patients with stage 4 metastatic disease and found that, in relationship to iNKT infiltration, survival at 5 years was 64% for iNKT^+^ and 35% for iNKT^−^ tumors (*p* = 0.007) [130]. Peng et al. tested fresh peripheral blood from 63 never-treated patients with gastric cancer and normal blood from 30 healthy control individuals using flow cytometry [131]. They found that the frequencies of CD3^+^CD56^+^ NKT-like cells were significantly lower in tumors (4.44%) compared to those in non-tumor tissues (7.20%). Molling et al. tested heparinized blood samples from 69 healthy subjects and 120 advanced cancer patients, none of whom had received chemotherapy or radiotherapy at the time of analysis, and found that cancer patients showed a 47% reduction in circulating NKT cells (*p* = 0.013) [132]. In another study also headed by Molling [133], circulating iNKT cells were evaluated in 47 patients with head and neck squamous cell carcinoma or HNSCC prior to radiotherapy. Clinical data obtained after a follow-up period of 31 months divided the patients into three separate groups with a small, intermediate, or large circulating iNKT cell fraction. These three groups were significantly associated with a decreased 3-year overall survival rate (39%, 75%, and 92%, respectively) and with a disease-specific survival rate (43%, 87%, and 92%, respectively). Klatka et al. [134] performed immunophenotyping on peripheral blood samples from 40 men and 10 women with laryngeal cancer in stages ranging from stage I to stage IV. The healthy control group consisted of 12 men and 3 women. Patients with advanced laryngeal cancer showed a significantly lower percentage of iNKT cells than controls (0.08% vs. 0.23%, *p* = 0.00046). These and many other demonstrations of low NKT levels correlating with poor outcomes in different kinds of cancer were summarized by Krijgsman et al. [129].

Alterations in the numbers and functions of NKT cells have also been associated with SSc, T1D, MS, and other autoimmune diseases [129,135,136,137,138,139,140,141,142,143]. Too many NKT cells could be the result of a strong immune response against microbial pathogens as a protective mechanism during the early phase of infections. High NKT frequency could be due to abnormally high NKT development in the thymus, an increased rate of basal proliferation, or an increased rate of survival in the periphery. In some autoimmune diseases in which a phospholipid was the target antigen, an increase in the number of NKT cells was observed [143]. This is because cardiolipin is among the reported lipid antigens recognized by CD1d [125,143]. Furthermore, increased numbers of CD3^−^CD56^+^ NK and CD3^+^CD56^+^ NKT cells were observed in patients with COPD compared to controls [141]. These numbers are an indication not only of immune dysfunction in COPD but also of the participation of NK and NKT cells in the pathogenesis of COPD. Elevated levels of circulating CD3^+^CD16^+^CD56^+^ cells were found in pregnant women and were associated with an increased rate of pregnancy and live birth in in vitro fertilization treatments [144].

The clinical significance of these seven major immune cell subpopulations and how they protect the body is summarized in Table 1.

These important subsets of immune cells are shown in Figure 2.

### 3.3. How the Innate and Adaptive Immune Systems Work Together to Protect the Body against Different Pathogens

To better understand the division of labor between NK and cytotoxic T cells in protecting the body against pathogens, we have summarized this information in the following seven steps (see Figure 3):In the absence of a functional innate and adaptive immune response, pathogens in infected cells or in their tissue environment grow exponentially. This exponential growth of the pathogens may result in serious inflammation, a cytokine storm, and death.Upon the detection of pathogens by antigen-presenting T-helper, NK cells, cytotoxic T cells and NKT cells become activated and produce significant amounts of IFN-γ within minutes. The job of interferon is to stop the spread of the pathogens before the NK cells and cytotoxic T cells arrive at the battleground.This production of IFN-γ by different cells inhibits pathogen replication, resulting in the flattening of the pathogen curve until the NK cells arrive at the site of the infection.The arrival of CD56^+^ and CD16^+^ NK cells at the infected area not only stops the replication of pathogens but also causes a significant decline in their numbers by killing the pathogens and the pathogen-infected cells.To finish the job, CD57^+^ and CD57^+^CD16^+^ cells, which are highly cytotoxic, are now summoned to join the operation in order to cleanse the body of the pathogen’s remnants, and, thus, the immune system prevails against infections.All these steps of pathogen elimination are part of the innate immune response in which the responding cytotoxic activity is carried out within minutes to hours. However, if the NK cell and its subpopulations fail to destroy the pathogen entirely, the immune system must call on additional resources.These are the complementary components of adaptive immunity, such as cytotoxic T cells, and their hybrids with CD57, the CD57^+^CD8^+^ cytotoxic T cells, will arrive at the pathogen-infected area within a few days of infection.

This is how the NK-cell and cytotoxic T-cell subpopulations respond during the early and late phases of infection by different microorganisms (see Figure 3).

## 4. Mechanisms by Which NK Cells Defend the Body against Microbes

One way in which NK cells defend against microbes is through their impressive array of activating and inhibitory receptors [25,28,182,183]. NK-cell receptors interact with a ligand that is expressed either by the microbe itself or by the infected host cell. Through these receptors, NK cells have been shown to recognize and eliminate cells infected with influenza, poxvirus, and other viruses [184,185,186,187]. While some ligands have already been identified, the exact corresponding microbial ligands for many NK-activating receptors are still largely unknown. The NK natural cytotoxicity receptor (NCR), receptor NKp46, for instance, can bind directly to *Pseudomonas aeruginosa* [188,189], *Nocardia farcinica*, and *Mycobacterium bovis* [190]. It is also through NKp46 that NK cells are activated by *Candida glabrata* [191]. Attempts are continually being made to identify other specific ligands for NK-activating receptors.

The interactions between the NK-cell receptors and the ligands on their targets determine the ability of the NK cells to destroy these harmful microorganisms or the cells that they have infected. Many viruses have, therefore, developed stratagems to affect or prevent detection, activation, and immune response. In HIV patients, for example, the expression of NK receptors is suppressed [28,192,193].

NK-cell receptors can also interact with ligands to regulate other NK receptors with regard to NK cells’ antiviral and antitumor functions. The tumor-necrosis-factor-related apoptosis-inducing ligand, or TRAIL, is found on cognate receptors expressed on target cells [194,195,196,197,198]. In rheumatoid arthritis, TRAIL signaling has been shown to limit pathology and inflammation [199,200]. However, TRAIL’s immunoregulatory role is somewhat of a double-edged sword, as its regulatory activities can lead to enhanced NK-cell-mediated T-cell killing [194].

The two processes described above show us that there are actually two main ways in which NK cells can kill and defend against harmful microbes: the direct antimicrobial cytotoxic pathway, in which NK cells come in contact with microbes and release cytotoxic granules to destroy them, and the indirect antimicrobial cytotoxic pathway, in which NK cells contact microbes and release cytokines, which then activate other immune cells to come and help destroy the dangerous microorganisms.

### 4.1. The Direct Antimicrobial Pathway

NK cells contain special materials called granules, which are filled with different highly potent effector molecules; this is why NK cells are called large granular lymphocytes. These molecules include a small antimicrobial protein called granulysin and a family of serine proteases called granzymes, as well as the pore-forming protein perforin [201,202]. When an NCR directly comes in contact with *Mycobacterium bovis*, the stimulation leads to activation of the protein kinases ERK, JNK, and p39 MAPK and direct cytotoxic activity in the form of the release of perforin and granulysin [33]. When NK cells encounter *Cryptococcus*, the NCR NKp30 and β-1 integrin cooperate in a process involving other immune components that end in the release of cytotoxic perforin-containing granules [38,203].

By releasing these granules, NK cells can kill viral-infected cells and cancer cells, defending the body against pathogens and tumors (see Figure 4).

### 4.2. The Indirect Antimicrobial Pathway

NK cells are involved in immunoregulation, and by producing different cytokines and mediators, they can shape adaptive immune responses [18,19].

As we have already stated, NK cells secrete several cytokines such as IFN-γ, TNF-β, IL-10, and IL-13. They also secrete granulocyte macrophage colony-stimulating factor (GM-CSF) and chemokines (CCL1, CCL2, CCL3, CCL4, CCL5, CXCL8) [16]. Because of this immunoregulatory function, they play a significant role in autoimmunity. NK cells are capable of an array of functions that range widely from their classic antitumor and antiviral cytotoxic effector functions to their critical immunoregulatory roles. For example, NK cells activate dendritic cells, macrophages, T cells, T-helper cells, and CD8^+^ cells. NK cells also stimulate plasma cells to produce antibodies [23]. These different roles and functions require NK cells to interact with other cells, as shown in Figure 5.

In the indirect pathway, pattern recognition receptors (PRRs) play a major role. NK cells are stimulated by bacterial pathogen-associated molecular patterns (PAMPs) into releasing IFN-γ in a Toll-like receptor 2 (TLR2)-dependent pathway [38]. When stimulated with *Aspergillus fumigatus*, TLR2 and TLR4 activate the eukaryotic transcription factor NF-κB [204].

Parasites can also induce this indirect response. NK cells can be activated by lipophosphoglycan from *Leishmania major*, resulting in enhanced production of IFN-γ and TNF-α and the nuclear translocation of NF-κB [205]. Lysates from *Plasmodium falciparum* also stimulate NK cells into producing IFN-γ [206].

Unfortunately, despite being small, single-celled microorganisms, bacterial pathogens are by no means simplistic or predictable, and some have evolved mechanisms to increase their probabilities for successful penetration, escaping detection and identification, survival, dissemination, and propagation. New bacterial strategies are constantly being recognized, and often bacteria will have multiple overlapping evasion mechanisms, making it more difficult for defensive immune cells to eliminate them [207,208,209].

## 5. How NK Cells and Cytotoxic T Cells Recognize and Eliminate Viral-Infected and Tumor Cells

The human body’s cells are exposed to and thus are affected by environmental factors. Whether they are nutritional, physical, chemical, pathogenic, or oncogenic, these factors can cause stress to the host cells, resulting in the induction of intrinsic and extrinsic cellular mechanisms aimed at counteracting the damage these factors inflict on the host [210]. The major extrinsic mechanism against cellular stress is the immune system, particularly NK cells.

### 5.1. How NK Cells Eliminate Target Cells

It has been well-documented that cells stressed by factors such as viral infection or early malignant transformation can be directly sensed or detected and recognized by NK cells, which then initiate cytotoxic action against the abnormal, damaged, or harmful cells. This cytotoxic action of the NK cell is divided into four main stages:

#### 5.1.1. Stage 1—Target Cell Identification

NK cells present a complex and variegated array of germline-encoded inhibitory and activating receptors that receive signals from MHC Class I (MHC-I) and Class I-like molecules, classical co-stimulatory ligands, and cytokines [211,212]. Normal healthy cells express on their surface MHC-I molecules, which act as ligands for the inhibitory receptors on NK cells and contribute to self-tolerance [23]. Virus-infected or tumor cells, however, suffer a decrease in their surface expression of MHC-I molecules, leading to lower or no inhibitory signal at all sent to the NK cells. Simultaneously, the cellular stress caused by the viral infection or tumor development activates the ligands for the activating receptors on the NK cells. Therefore, if an NK cell finds a damaged or abnormal cell with little or no MHC-I molecules and an upregulated activating ligand, it will not receive an inhibitory signal, but will instead receive an activating signal that identifies the subject cell as a target for elimination [23]. The activating signals induce the accumulation and movement of the granules containing perforin and granzymes through microtubules formed between the NK cell and the target cell [213]. These signals also lead to the transcription of cytokine and chemokine genes [23]. This communication between the NK cell and target cells is very important because it ensures that only harmful cells are destroyed and not healthy cells (see Figure 6). In sum, if an NK cell receives an inhibitory signal from an MHC-I molecule on a suspect cell’s surface, the NK cell will recognize the target cell as “self”, and it will not destroy that cell. If the NK cell does not receive the inhibitory signal but receives an activating signal instead, it will destroy the target cell [20,23]. This is called the “missing self” hypothesis.

#### 5.1.2. Stage 2—Formation of Immunological Synapse

If the NK receptors have determined that the target is a harmful cell, an immunological synapse is formed between the NK cell and the target cell. The microtubule organizing center (MTOC) and secretory lysosome polarize towards the immunological synapse [201]. The NK cell’s secretory lysosome docks or fuses with the target cell’s membrane, forming a bridge or pathway between the two cells through which the NK cell’s special molecules can pass [214].

#### 5.1.3. Stage 3—NK-Cell-Induced Target Cell Death

The next phase of the NK cell’s cytotoxic action is the actual destruction of pathogen-infected cells, cancer cells, cells modified by hazardous chemicals, or other stressed cells. This process, which began with immunological synapse formation, is called the degranulation process or release of the granules. Following the docking process, the launched granules release perforins that cause pore formation in the surface membrane of the target cell. The granules also release granzymes and other enzymes, such as serine proteases. Using the pores opened by perforin, these enzymes gain entry into the harmful cells, where they activate caspase molecules that induce the death of the target cell, also known as apoptosis [33,72,201,215].

#### 5.1.4. Stage 4—Post-Apoptosis Detachment of the Killer Cell

Once the target cell starts to die or commences apoptosis, its cytoskeleton undergoes what is called apoptotic contraction. The killer cell senses what is going on through the immunological synapse, and consequently it breaks its connection with the target cell, releasing the dying cell so that the killer can go on to target other cells in what is known as serial killing [216,217].

These stages of the cytotoxic process are elegantly discussed in articles by Paul and Lai [23] and Sanchez et al. [216,217].

### 5.2. How CD8^+^ T Cells Eliminate Target Cells

CD8^+^ T cells are essential in defending the host body against pathogens. Unlike an NK cell, which needs an MHC-I molecule to signal to it that a cell is harmless, a cytotoxic CD8^+^ T cell needs an MHC-I molecule to signal to it that a cell needs to be destroyed. This is because a CD8^+^ T cell identifies harmful cells by recognizing an infecting pathogen’s foreign peptides that have been transported to the infected cell’s surface bound to MHC-I molecules. This process is called licensing. Once thus triggered, the CD8^+^ T cell is now “licensed” to eliminate any other cells it encounters that have the same MHC-I/peptide complex [218].

Once the CD8^+^ T cell or cytotoxic T cell (CTL) has recognized a specific antigen on the cellular target, the CTL assembles an immunological synapse where it arranges the tools necessary for eliminating the harmful cell. These lysing tools are stored in lysosome-related organelles (LROs), which undergo exocytosis in response to the CTL’s recognition of the pathogen’s antigen [114]. Recent discoveries have revealed that these LROs include not just the well-known lytic granules but also multivesicular bodies carrying the ligand FasL, and supramolecular attack particles or SMAPs [114]. This means that CD8^+^ T cells have three pathways to CTL-mediated elimination of harmful cells.

The first pathway is the granular pathway. Lytic granules or LGs are composed of an outer membrane that contains a battery of serine proteases with different substrate specificities, granzymes, perforin (a protein that can perforate the target cell’s membrane, hence the name), and granulysin, a saposin-like membrane-disrupting protein, all packed together [114]. When a target cell is successfully identified, granules are launched, carrying their special cargos. Synapses are established, perforin makes pores in the target cell’s membrane, and the other cargo molecules pass through, causing cellular cleavage, destruction, and cell death or apoptosis (see Figure 4).

In the second pathway, apoptosis occurs through an enzyme caspase-dependent manner involving the FS-7 associated surface antigen, or Fas for short. Fas, also known as CD95, APO-1, TNFRSF6, or the death receptor, is a type I transmembrane protein containing a death domain in its cytoplasmic region, and it is essential for the induction of apoptosis [215]. It is expressed on the surface of target cells. CTLs express on their surface a matching ligand to Fas, called the FasL. FasL is a type II transmembrane protein that is upregulated at the cell surface in response to target cell recognition [114]. It is stored in multivesicular bodies (MVBs) that are similar but larger (300–700 nm) than the regular granules that carry granzymes and perforin (<300 nm). Upon target cell recognition, the multivesicular bodies containing the FasL polarize towards the centrosome and undergo fusion with the synaptic membrane, releasing the FasL [114]. When the FasL from the CTL comes into contact with the Fas on the target cell, this activates procaspase-8, caspase-8, and effector caspase-3 in a process that ultimately results in the apoptosis or cell death of the tumor cells or viral-infected cells (Figure 7) [215].

The third pathway involving the recently discovered SMAPs is an unconventional mechanism of CTL-mediated cytotoxicity [102]. SMAPs are cytotoxic multiprotein complexes [219]. They are stored in multicore granules and are thought to be autonomously cytotoxic. It was observed that after CTLs were removed from the target, SMAPs were left behind.

In the face of all these different pathways towards their elimination by both NK cells and CD8^+^ CTLs, abnormal cells, such as cancer cells, in order to survive, need to evolve mechanisms to avoid elimination, and one such mechanism is the shedding of the MHC-I molecule. These molecules are not essential for cell survival, and by shedding them, abnormal or enemy cells can avoid elimination by CD8^+^ T cells by not having an MHC-I/peptide complex that the “licensed” CD8^+^ T cells can recognize [220].

Unfortunately for harmful, infected, or malignant cells, while this mechanism may work on the CD8^+^ T cell, it has quite the opposite effect on its partner, the NK cell. Remember that an NK cell needs to find an MHC-I molecule on a suspect cell’s surface in order to recognize it as harmless. The downregulation or even absence of MHC-I molecules on infected cells and tumor cells will tell the NK cell that this is an abnormal cell that must be destroyed. In complementary fashion, the NK cell can eliminate what the CD8^+^ T cell has missed.

Because of this capability and efficiency of the NK cell to kill target cells, some call the NK cell the serial killer of tumor cells [221]. Interestingly, NK cells switch from the fast release of perforin- and granzyme-mediated killing during the initial killing events to a slow death receptor-mediated killing during subsequent contacts with tumor cells [222]. These dual mechanisms of cancer-cell killing by NK cells are differentially regulated during NK serial killings. It seems that the slower death receptor-mediated apoptosis or killing of tumor cells serves as a backup mechanism when the granzyme and perforin contents of the granules have been exhausted.

## 6. Potential Clinical and Therapeutic Applications of NK and CD8^+^ Cells

Considering the various components of the immune system, all the different kinds of cells involved, and the inter-communication and collaboration between them, any factor that affects one component may also affect the other components with which it interacts [223]. It is possible for these triggers and conditions to impact the production, function, and efficiency of the different specific subpopulations of NK cells and CD8^+^ T cells, so that the measurement of these subpopulation levels and percentages could give very helpful information about the many diseases that affect human populations [224,225,226,227,228,229].

Alterations in NK-cell and CD8^+^ T-cell subpopulations have been observed in the following different major disorders:Viral infections including COVID-19 [51,102,103,104,105,110,129,135,136,137,138,140,230];Long COVID due to viral persistence [101];Myalgic encephalomyelitis/chronic fatigue syndrome (ME/CFS) [224];Cancer [23,58,60,61,62,63,64,65,66,67,68,69,70,109,110,116,129,143,178,197,225];Autoimmune disease including celiac disease: early stages and full-blown [48,57,60,61,74,75,76,77,78,79,80,81,82,83,84,85,86,87,90,91,92,93,106,115,125,126,129,135,136,137,138,139,143,146,231,232,233,234,235];Toxic chemical exposure [235,236,237];Chronic alcoholism due to neoantigen formation [238,239];Chronic pulmonary disease [123,140,141];Chronic severe pain [240,241];Transplantation [124,166,167,242];Acute physical stress [59,240,241,242,243,244,245,246];Neuropsychiatric disorders [247,248,249];Neurodegenerative disorders [248,249,250,251,252];Early pregnancy loss [52,144,253];Immune deficiency [20,153,158,223];Broken essential barriers [254,255];Autism [256,257,258].

All the above disorders and their associations with abnormality of NK and CD8+ T cells encourage further investigation of different lymphocytes, particularly the deficiency or over-activity of NK and CD8+ T cells, in designing therapeutic protocols for not only dealing with pathogens and cancers but also many disorders associated with immunodysregulation.

In fact, researchers are working very hard to make NK and NKT cells smarter in order to improve cell therapy, as well as exploring approaches to activating and expanding NK and NKT cells through methods such as adoptive transfer of cells in humans [259]. In addition to various protocols for the enhancement of NK- and cytotoxic T-cell activity using biological response modifiers, fundamental progress has been made in the genetic and therapeutic manipulation of NK and other immune cells in cancer, infectious diseases, and autoimmune disorders:One such approach is the use of NK cells for cancer immunotherapy. Advances in viral transduction and electroporation now allow for detailed characterization of genetically modified NK cells and provide a better understanding of how these cells can be used clinically to optimize their capacity to induce tumor regression in vivo through approaches such as improving NK-cell persistence via autocrine IL-2 and IL-15 stimulation, enhancing tumor targeting by silencing inhibitory NK-cell receptors such as NKG2A, and redirecting tumor killing via chimeric antigen receptors [260]. A review by Vishwashrao et al. focused on the use of CARs, BiKEs, and TriKEs to engineer NK cells for tumor immunotherapy. CARs are chimeric antigen receptors that can improve the cytotoxicity of effector cells by specifically redirecting their destructive capability towards a defined target antigen on a tumor or transformed cell. The addition of CAR expression to existing NK-cell-activating receptors could enhance NK cells’ ability to eliminate targeted tumor cells, especially in solid tumors often resistant to NK-cell-mediated killing. BiKEs and TriKEs are bispecific and trispecific killer engagers that can further enhance cytotoxic killing and cytokine production [261]. In a very recent review, Maia et al. described how genetic modifications can enhance NK cells’ tumor-targeting capability, cytotoxicity, persistence, and tumor infiltration, and also prevent exhaustion, as well as detailed both non-viral (electroporation, lipid nanoparticles, lipofection, DNA transposons) and viral (lentivirus, gamma retrovirus, adeno-associated virus) technologies for genetically modifying NK cells [262].The second approach is therapeutic manipulation of NK cells in infectious diseases. The anticancer effect of NK cells is already being investigated in multiple clinical trials, but relatively little is known about the therapeutic utilization of NK cells in patients suffering from infectious complications. In vivo data clearly show that NK cells are active against viral, bacterial, and fungal pathogens, and the adoptive transfer of NK cells seems to be a promising methodology. However, although animal studies have been encouraging, it should be remembered that purposeful activation of the host immune system has been shown to have severe complications in humans. Again, more research is needed to fully characterize the optimal patient population, the best time point of the therapy, and the best approach to generating NK cells for immunotherapy [21].The third approach is genetic manipulation of NKT cells in autoimmunity. Several autoimmune diseases show defects in the functionality of NKT cells. Attempts have been made to utilize molecules’ ability to activate NKT cells. For instance, in mice, injection of the glycolipid α-GalCer led to upregulation of iNKT cells, which, in turn, led to the activation of NK cells through IFN-γ produced by the iNKT cells. However, further research is needed as repeated injections seemed to exacerbate some autoimmune diseases, and tests in humans seemed to be less effective than in mice. On the other hand, increasing the number of iNKT cells through adoptive transfer seemed to be safe and well-tolerated in human trials. Another method tested is the promotion of self-ligands that activate iNKT cells [263].A fourth approach is the therapeutic transfer of ex vivo expanded ILC2s to induce tumor cell death. In a major breakthrough reported by Li et al. [264], the researchers found that as a member of the cytolytic immune effector cell family, human ILC2s secrete granzymes that lyse tumor cells by inducing pyroptosis and/or apoptosis of tumor cells. Unlike CAR T-cell therapy, which requires specific characteristics, ILC2 can be isolated from healthy donors, which is a huge advantage. Thus, in their experiments, Li et al. isolated cells from the blood and expanded them by 2000-fold in culture. The expanded ILC2s were exogenously administered to patients with leukemia, and to solid tumor models in vitro and in vivo. Their findings suggested the potential usefulness of ex vivo-expanded human ILC2s as an adoptive cell strategy for cancer immunotherapy.

All of these novel methods have been used to genetically manipulate, activate, or expand human NK, NKT, and other immune cells ex vivo for adoptive transfer in humans.

Additional novel strategies are used to manipulate NK-cell function through the use of antibodies targeting KIR, NKG2A, and NCR as major inhibitory checkpoints [265,266], to treat diseases other than tumors, including infections, inflammation, and autoimmune diseases.

## 7. Summary and Conclusions

In most articles, NK cells are distinguished into two subsets according to their surface expression of CD56 and CD16, but in this review, we have described the functions of four different subpopulations of NK cells: CD56^+^CD16^−^, CD56^−^CD16^+^, CD57^+^, and CD57^+^CD16^+^ NK cells, which are major components of innate immunity.

The adaptive counterparts of the innate NK cells are the CD8^+^ and CD57^+^CD8^+^ cytotoxic T cells, the two main effectors of the adaptive immune system.

Finally, we have discussed the importance of CD3^+^CD16^+^CD56^+^ NKT cells. Known as the bridge between innate and adaptive immunity, NKT cells play a greater role in immunoregulation than in the actual killing of pathogens.

We have shown how these seven important subpopulations of NK and CD8^+^ T cells fight many different diseases with their own diverse and specific functions and capabilities. The studies discussed in this article about these seven lymphocytes and briefly about ILC2s provide further insights into the anticancer and antimicrobial functions of NK cells, CD8^+^ cells, and ILC2s, with an emphasis on their characteristics and their clinical significance. A better understanding of the mechanisms by which NK-cell, CD8-cell, and ILC2 subpopulations destroy their target cells will help researchers and clinicians to optimize their defensive functions and properties against some life-threatening diseases.

To achieve this better understanding, further investigations should be performed on the defensive strategies of pathogens and tumor cells against NK cells, CD8 cells, and their subpopulations so as to find ways in which these protective immune cells can counteract the evasive tactics of their target cells. More research is also needed to expand the list of molecular targets that could be modulated in NK and CD8^+^ cells through the use of biologics and other factors to devise therapeutic protocols, ameliorate suffering, and improve human health.

## Figures and Tables

**Figure 1 microorganisms-12-00230-f001:**
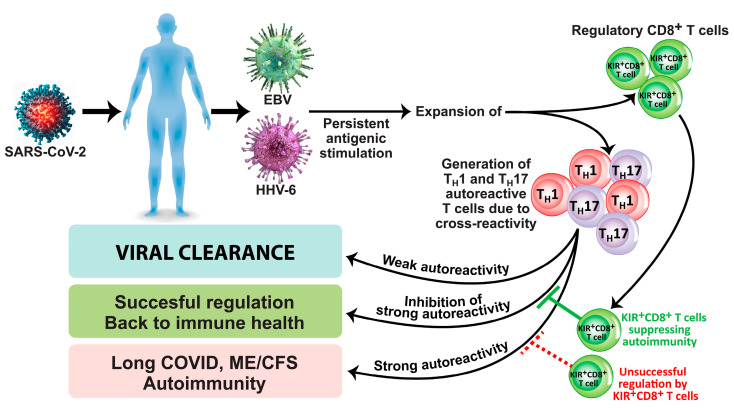
Induction of long COVID, ME/CFS, and autoimmunity due to viral infection/reactivation in the absence of successful regulation by KIR^+^CD8^+^ regulatory T cells.

**Figure 2 microorganisms-12-00230-f002:**
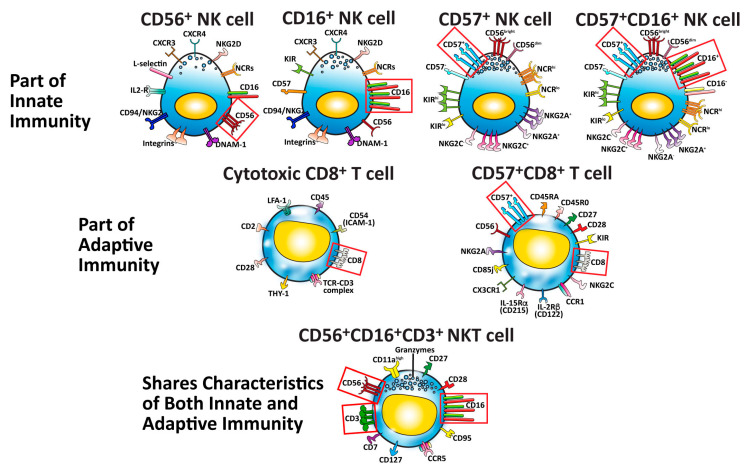
Seven major cells that protect our body against pathogens, cancers, and neo-antigens. The identifying cluster differentiation (CD) markers for each specific cell type are outlined in red boxes.

**Figure 3 microorganisms-12-00230-f003:**
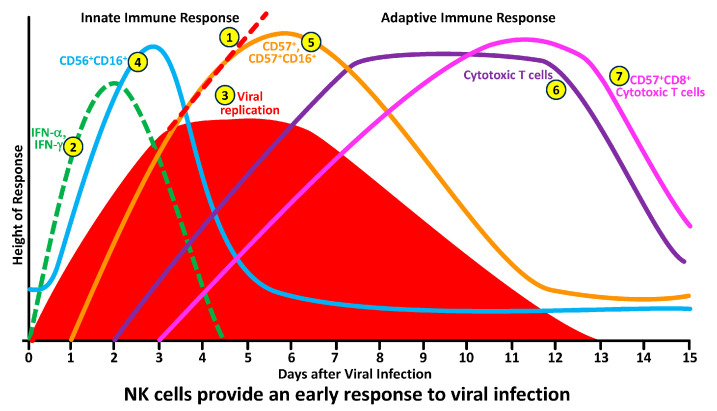
How NK and cytotoxic T-cell subpopulations respond during the early and late phases of infection.

**Figure 4 microorganisms-12-00230-f004:**
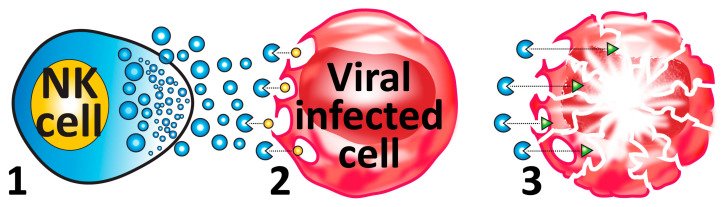
NK granular release. (**1**) The activated NK cell releases the granules (blue molecules). (**2**) The granules release perforin (yellow molecules), which blasts holes in the surface membrane of the viral-infected or cancer cell. (**3**) The granules then release the granulysin (green molecules), which enters the cell through the holes made by the perforin and induces apoptosis or cell death.

**Figure 5 microorganisms-12-00230-f005:**
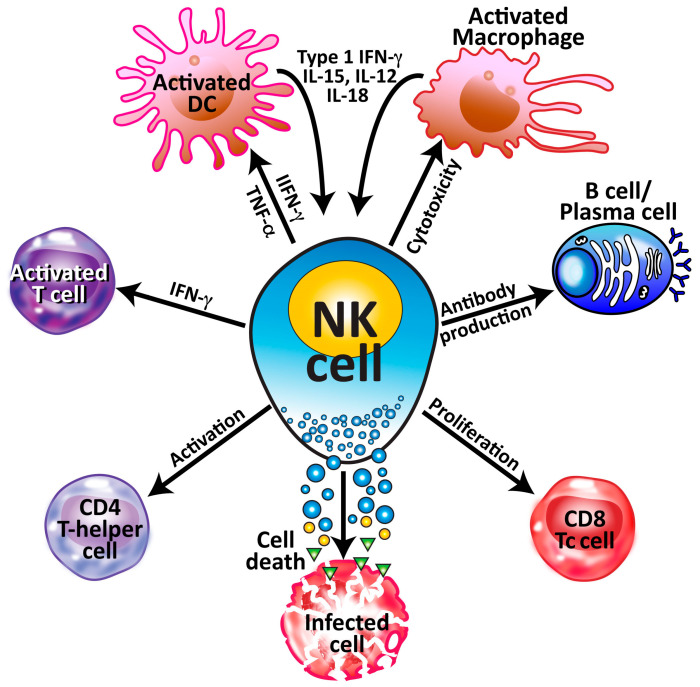
The multiple interactions of NK cells with other cells of the immune system. NK cells perform various functions and roles that require them to interact with different cells in different ways, from shaping immune responses through the production of cytokines, to the outright killing or destroying of infected cells or cancer cells through the release of granules.

**Figure 6 microorganisms-12-00230-f006:**
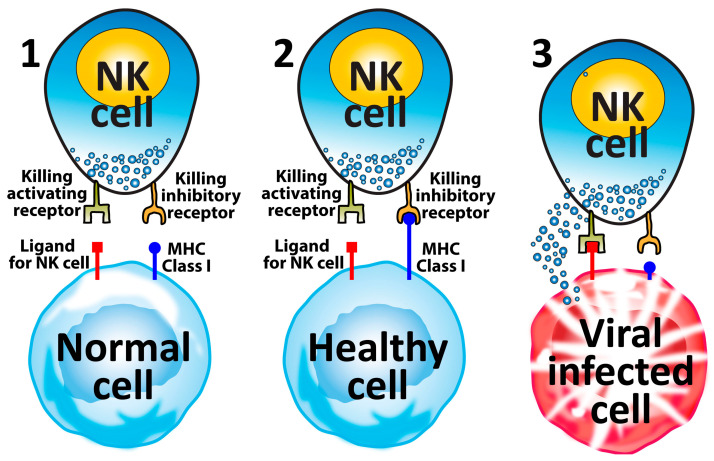
Why NK cells kill viral-infected cells and tumor cells, but not healthy cells. (**1**) Two different receptors on NK cells and the ligands on healthy cells ensure that NK cells do not attack normal cells. (**2**) Healthy cells express a significant level of MHC Class I, which engages with the inhibitory receptor. Because the ligand for NK cells does not engage the activating receptor, the NK does not kill the healthy cell. (**3**) In viral-infected and cancer cells, MHC Class I is significantly reduced, preventing the inhibitory receptor from engaging with it. But the activating receptor on the NK cell binds strongly to the ligand for NK cells on the viral-infected/cancer cell, causing the release of destructive granules from the NK cell, resulting in the killing of the viral-infected cell or tumor cell (see also Figure 3 and Figure 4).

**Figure 7 microorganisms-12-00230-f007:**
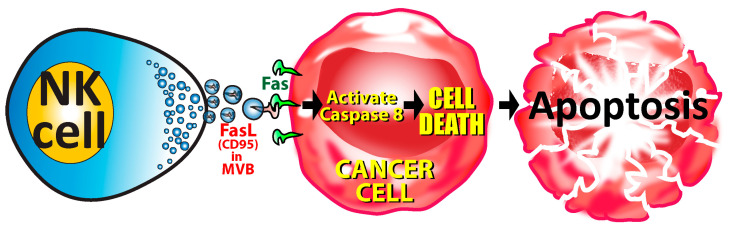
The caspase-mediated pathway to apoptosis. The binding of the CD95 Fas ligand (FasL) from the NK cell to the Fas on the tumor cell initiates a chain of events that activates Caspase 8, ultimately leading to the death or apoptosis of the cancer cell.

**Table 1 microorganisms-12-00230-t001:** Characteristics of human NK cells and CD8^+^ T cells and their clinical significance.

Immunophenotype	Cytotoxic Activity	Clinical Significance	References
CD56^bright(+)^CD16^−^ NK Cell	Weakly cytotoxic Produces limited amounts of cytotoxic molecules	Immunoregulation through production of many cytokines and chemokines	[145,146,147,148,149,150,151,152]
CD56^−^CD16^bright(+)^ NK Cell	Naturally strongly cytotoxic Produces significant amounts of cytotoxic molecules	Protection against microorganisms and cancerous cells	[153,154,155,156,157,158,159,160,161]
CD3^−^CD57^bright(+)^ NK Cell	High cytotoxic potential as a result of producing various cytotoxic molecules	Long-lived cells Immunoprotection against pathogens, especially during aging Deletion of autoreactive T cells Prevention of some autoimmune diseases	[153,154,155,156,157,158,159,160,161,162,163]
CD57^bright(+)^ CD16^bright(+)^ NK Cell	Highly cytotoxic Produces significant amounts ofcytotoxic molecules	The most potent cells for combating acute and chronic infections	[69,71,162,163,164,165,166,167,168]
CD8^+^ Cytotoxic T Cell	After activation by APC, becomes highly cytotoxic Produces different cytotoxic molecules	Immune response to bacterial/viral infections and cancer	[113,169,170,171,172,173,174,175,176,177,178]
CD8^+^CD57^+^ T Cell	Highly cytotoxic Produces significant amounts of cytotoxic molecules	Mediators of adaptive immunity Associated with chronic immune activation Age-related changes in the immune system status	[59,60,61,81,114,179,180,181]
CD3^+^CD56^+^CD16^+^ T Cell	Moderately cytotoxic Plays a significant role in immunoregulation	Serves as bridge between innate and adaptive immunity Immune protection against microbial pathogens and cancer Control of autoimmune diseases	[129,136,137,138,139,141,142]

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
