# Peer review of "Natural Killer Cells and Cytotoxic T Cells: Complementary Partners against Microorganisms and Cancer"

_microorganisms, 2024, doi:10.3390/microorganisms12010230_

Round 1
Reviewer 1 Report (Previous Reviewer 1)
Comments and Suggestions for Authors
The paper is improved. Writing style is somewhat casual in places, but acceptable.
Author Response
Thank you for your help in improving our article.
Aristo Vojdani, Corresponding Author
Reviewer 2 Report (New Reviewer)
Comments and Suggestions for Authors
The authors emphasize with this Review the relationship between NK and TCD8 cells, defining their role in combacting microrganisms and cancer.
In general the manuscript is well written and focused the attention on two important cells population: Natural Killer and TCD8 cells. I appreciate a lot the figures and tables. Tables are well organized and precise.
Here my comments and some suggestions for the authors:
In the Introduction part, the authors should mentioned also T cells with a particular focus on TCD8 cells.
Check line 57 there is a doble bracket “e [[16,17]. If the…”
NK cells today are also considered as a part of ILCs, so I kindly suggest the authors to mentioned also Hergen Spitz, that is the milestone of ILCs discovery as well as Garofalo et al Cancer 2023 and Cristiani et al Frontier in Immunology 2022. This part could better focused the NK cells population and render your manuscript more updated.
Lines 111-112: I don’t think is necessary to include this part on mouse NK cells, also because the Review is focused on human NK and T cells.
Author Response
Please see the attachment.

This manuscript is a resubmission of an earlier submission. The following is a list of the peer review reports and author responses from that submission.
Round 1
Reviewer 1 Report
Comments and Suggestions for Authors
This review paper by Vojdani and colleagues discusses the roles of NK and CD8+ T cells. The review is beautifully illustrated. The style of writing is entertaining but unusual for a scientific manuscript. There is a heavy emphasis on surface markers to identify sub-sets of NK cells, and a section devoted to flow cytometry which I don’t think is pertinent to the review. Since the title of the manuscript is “the front line against microorganisms” CD8 T cells do not really fit, as they point out that themselves that these cells take time to be selected and expand in response to antigens that are presented by MHC on antigen presenting cells.
They do not deal with other lymphocytes such as IEL.
While they claim that the NK cells directly kill a variety of microbes, there are no citations that show that that NK cells are critical for combating the infections caused by those microbes. Is there such evidence?
Is it not true that pathogens by definition have figured out how to evade innate immune defenses?
The authors say that gamma interferon made by NK cells plays a large role in anti-viral defense, but it is really alpha- interferons that are crucial and gamma interferon has rather weak anti-viral activity.
On line 632 they say NKT cells protect against antigens, but they protect against microbes not antigens.
Beginning on line 560 they list many diseases and syndromes associated with alterations in NK and CD8+ T cells and imply that it is important to measure those cells in patients. However, there is no citations to show that this information either aids in diagnosis or treatment of those conditions.
Reviewer 2 Report
Comments and Suggestions for Authors
The manuscript Natural killer cells and cytotoxic T cells: The front line against microorganisms by Aristo Vojdani et al., makes an interesting update of the phenotypic and functional characteristics based on results from various groups and based on the bibliography. The approach proposed by the authors in the current conditions is of little delicacy and encourages violence, although simple drawings and figures are proposed to try to understand the relevance and complexity of microbicidal mechanisms, it is presented in very bad taste in conditions of violence. world. Under these conditions, it is suggested that the manuscript be rejected, however, the authors are invited to reconsider the work and present it clearly but omitting information outside the scientific context, talk about Superhero Team-Up, heroes, missiles, Figure 3 and 4.NK missile attack etc. As well as graphic bombardment? The front line (in the title) in this context then refers to the war front ?.
It is regrettable that in the interest of presenting the most understandable information possible it has been trivialized with a bellicose character for which this scientific forum should not lend itself
Round 2
Reviewer 2 Report
Comments and Suggestions for Authors
3.
